# Does Repeated Dosing of Intravenous Ferric Carboxymaltose Alleviate Symptoms of Restless Legs Syndrome?

**DOI:** 10.3390/jcm11061673

**Published:** 2022-03-17

**Authors:** Hea Ree Park, Su Jung Choi, Eun Yeon Joo

**Affiliations:** 1Department of Neurology, Inje University College of Medicine, Ilsan Paik Hospital, Goyang 10380, Korea; okokree@gmail.com; 2Graduate School of Clinical Nursing Science, Sungkyunkwan University, Seoul 04514, Korea; sujungchoi@hanmail.net; 3Department of Neurology, Neuroscience Center, Samsung Medical Center, Sungkyunkwan University School of Medicine, Seoul 16419, Korea

**Keywords:** RLS, ferric carboxymaltose, iron, intravenous

## Abstract

Background: Several studies have reported the efficacy of intravenous (IV) iron for patients with restless legs syndrome (RLS), but little is known about the efficacy or safety of repeated IV iron treatment. The aim of this study was to evaluate the effectiveness of repeated doses of IV ferric carboxymaltose (FCM) in treating RLS symptoms. Methods: Patients who received FCM more than twice for RLS from April 2016 to January 2020 were retrospectively reviewed. Patients who had shown positive response to initial IV FCM re-visited the clinic when their symptoms returned, and received repeated IV FCM (1000 mg). Blood iron panels were measured before initial and repeated IV FCM. We defined ‘responders’ as patients with a greater than 40% decrease in International Restless Legs Study Group Severity Scale (IRLS) compared with pre-treatment levels. Results: A total of 42 patients, including 26 with primary RLS, 5 with gastrectomy, and 11 with anemia, completed the evaluation. Patients received IV FCM infusion 2–4 times. A total of 21 of 26 (80.8%) primary cases of RLS, 4 of 5 (80.0%) patients with a history of gastrectomy, and 9 of 11 (81.8%) patients with anemia responded to repeated FCM treatment. Serum ferritin levels of patients with primary RLS were higher before the second treatment than the baseline levels. There were no serious adverse events observed in the study. Conclusions: Repeated IV FCM for recurring symptoms is an effective treatment for primary RLS and RLS associated with iron deficiency. Serum ferritin might not be a reliable factor to monitor the sustained effects of IV iron for RLS.

## 1. Introduction

Brain iron deficiency plays a key role in the pathophysiology of restless legs syndrome (RLS) [1,2,3]. In recent years, an increased number of clinical studies reported the role of intravenous (IV) iron treatment in RLS. IV iron bypasses the gastrointestinal-based regulation of oral iron absorption, and, thus, provides a rapid (4–8 weeks) treatment response compared with that of oral iron (2–6 months), with low gastrointestinal discomfort [4,5,6]. Among currently available IV formulations, ferric carboxy maltose (FCM) is a stable compound that releases iron more slowly than faster-release formula (iron sucrose and iron gluconate), and produces less toxic, free or labile iron [7,8]. Its properties permit the administration of large doses in a single brief session. Until now, three randomized, placebo-controlled, double-blind studies, and several observational studies of IV ferric carboxymaltose (FCM, total dose of 1000 mg) demonstrated significant improvement in RLS symptoms [4,9,10,11,12,13,14]. Based on this evidence, the International RLS Study Group (IRLSSG) task force recently recommended IV FCM for adult patients with moderate-to-severe RLS, serum ferritin ≤100 μg/L, and percent transferrin saturation (%TSAT) <45 as an alternative to oral iron supplementation [8].

Nevertheless, evidence is inadequate to support the duration of IV FCM treatment effect and the efficacy of repeated IV iron doses for patients with recurring RLS symptoms. The two studies from the follow-up phase of randomized trials of IV FCM mentioned above demonstrated persistent treatment response in 60% of responders at 20 weeks, and 37.5% at 30 weeks [4,13]. A single open-label study that used IV dextran reported that the duration of treatment effect in six responders ranged from 3 to 36 months with a mean duration of 11.3 months [5]. A small open-label study suggested that treatment with repeated IV iron maintained efficacy [15]. However, the number of subjects (only five subjects treated with repeated IV iron) was not adequate to assess the treatment efficacy. Moreover, IV iron sucrose used in a previous study showed insufficient outcomes in randomized trials [16,17]. No clinical studies have investigated the safety and efficacy of repeated IV FCM treatments in patients with RLS. The IRLSSG clinical practice guidelines underscore the need for further studies to investigate the results of repeated IV iron treatment [4].

To explore the benefit of repeated IV FCM treatment for patients with RLS who suffered from the return of RLS symptoms after a successful initial treatment with IV FCM, we analyzed iron profiles and treatment response following repeated IV FCM administration.

## 2. Materials and Methods

### 2.1. Subjects

We retrospectively evaluated a group of consecutive patients aged ≥18 years and diagnosed with RLS, who received IV FCM treatment more than twice at our sleep clinic between April 2016 and January 2020. A diagnosis of RLS was established by neurologists after an interview and physical examination in the sleep clinic based on updated IRLSSG diagnostic criteria [18]. We included patients who showed an initial positive clinical response to IV FCM, and met the iron status criteria defined for evidence-based treatment recommendations, i.e., serum ferritin ≤300 μg/L and %TSAT ≤ 45 [8]. We defined an initial positive clinical response to IV FCM as a more than 40% decrease in International Restless Legs Study Group Severity Scale (IRLS), following previous studies of IV FCM treatment [4,12,13,19]. We excluded patients who were lost to follow-up or failed to undergo an objective assessment (i.e., IRLS) after initial and repeated doses of IV FCM (i.e., IRLS change), as well as those who had high ferrin (>300 μg/L) or %TSAT over 45. Detailed information about comorbid diseases and medication history was obtained. Subjects with anemia and history of gastrectomy were analyzed separately from subjects with primary RLS.

This study was approved and monitored by the Institutional Review Board of Samsung Medical Center (IRB No. 2017-12-082-001), and the need for informed consent was waived. This study was conducted in compliance with the Declaration of Helsinki and Good Clinical Practice guidelines.

### 2.2. Clinical Data and Blood Tests

Data were collected using the following questionnaires: Pittsburgh Sleep Quality Index (PSQI), Insomnia Severity Index (ISI), IRLS, and Hospital Anxiety Depression Scale (HADS). We evaluated patients with RLS, including disease duration, family history of RLS, and current use of RLS medication (dopamine agonists with or without alpha-2-delta ligands). Morning fasting blood samples were obtained before initial and repeated IV FCM treatment, and analyzed for complete blood cell count, serum iron, ferritin, and total iron-binding capacity (TIBC).

### 2.3. Repeated FCM Treatment Protocol and Outcome Measures

During the initial FCM treatment, all subjects received a single IV dose of 1000 mg FCM in 100 mL of normal saline solution delivered over 15 min. Treatment response was assessed based on the difference between baseline IRLS, determined on the day of FCM treatment, and post-treatment IRLS, obtained four weeks after treatment. Patients with a greater than 40% decrease in IRLS from baseline were considered “responders”. These classification criteria are consistent with those of previous studies of IV FCM treatment [4,12,13]. In patients taking medications for RLS symptoms, IV FCM was administered without modification of previous medication, and these patients were on a stable dosage of medications throughout the study duration.

In the case of responders to initial FCM treatment, we instructed patients to re-visit the sleep clinic when their symptoms of RLS returned, in order to receive repeated IV FCM. We instructed them not to add medications for relieving leg discomfort or increase the dosage of their medications before they re-visit the sleep clinic, because we choose repeated FCM injection as the most effective first-line treatment for these cases. We did not encourage these responders to visit the sleep clinic regularly to follow any changes in leg discomfort or iron status until their leg discomfort returned. Thus, the time interval between initial treatment and re-evaluation of relapsed symptoms for repeated treatment was uneven, depending on the patients. All subjects underwent re-evaluation of IRLS and blood sampling before a repeated dose of IV FCM. We did not administer repeated IV FCM for patients who carried a high ferritin level (>300 mg/L) or a high TSAT% (>45%) in the follow-up blood tests, following the IRLSSG clinical practice guidelines [8]. The dose and method of repeated FCM administration were the same as those of the initial treatment. Post-treatment IRLS were obtained four weeks after administration of repeated IV FCM, and the classification of treatment response to repeated IV FCM was the same as the initial treatment.

### 2.4. Statistical Analysis

All statistical analyses were performed using SPSS version 18.0 for Windows (SPSS Inc., Chicago, IL, USA). Baseline characteristics and the proportion of responders were assessed separately among the three groups: primary RLS, secondary RLS with a history of gastrectomy, and those with RLS and anemia. One-way ANOVA tests for continuous variables, and chi-square tests and Pearson’s chi-square test for categorical variables, were used to compare the baseline characteristics and treatment response among the three groups. The changes in IRLS and iron panel profile after the first and repeated-dose IV FCM were compared in the primary RLS group using a paired *t*-test because all data used for this comparison followed the normal distribution. Iron characteristics were analyzed separately for males and females. Statistical significance was defined as *p* < 0.05.

## 3. Results

Figure 1 summarizes patient enrollment and classification. Among 324 patients who underwent IV FCM treatment, repeated IV FCM was administered to 65 subjects, 54 of whom met the study enrollment criteria. Overall, 12 patients dropped out, leaving 42 with complete evaluations. The 42 study completers comprised 26 with primary and 16 with secondary RLS. The secondary RLS group included 5 patients with gastrectomy and 11 with anemia; they suffered from moderate to severe RLS (mean baseline IRLS 30.0 ± 7.7). Other patient clinical characteristics are summarized in Table 1. Blood iron panels including hemoglobin, ferritin, %TSAT were significantly different among groups, and primary RLS patients’ hemoglobin and % TSAT were higher than other patients. All patients responded to initial treatment with IV FCM. Two patients experienced a transient headache after IV FCM infusion, whereas the others did not report any side effects (e.g., extravasations, hypersensitivity reactions) of IV FCM treatment.

### 3.1. Treatment Response to Repeated IV FCM in Patients with Primary and Secondary RLS

Table 2 presents the treatment response of primary and secondary RLS completers to repeated iron treatment. In the primary RLS group, 24 of 26 (92.3%) patients were treated with IV FCM twice, and 2 of 26 (7.7%) patients were treated three times during the study period (45 months). Among the five patients with a history of gastrectomy, four patients were administered IV FCM twice and one patient was treated three times. Overall, 7 of 11 (63.6%) patients with anemia were treated with FCM twice, and two patients (18.2%) were treated three times, with the remaining two cases administered four times. The time interval between the first and second FCM administration was 14.5 ± 8.0 months for patients with primary RLS, 10.6 ± 3.2 months for patients with a history of gastrectomy, and 11.6 ± 3.0 months for patients with anemia. Statistically, there was no significant difference in the magnitude of this time interval between the three groups (*p* = 0.313). In the primary RLS completer group, 21 of 26 subjects (80.8%) responded to the second IV FCM (first repeat) treatment. In the secondary RLS group, 4 of 5 (80.0%) subjects with a history of gastrectomy and 9 of 11 (81.8%) subjects with anemia responded to the second IV FCM treatment. For patients who were administered IV FCM treatment more than three times, their mean intervals between second and third treatment was 10.3 ± 4.3 months, and between third and fourth treatment was 11.3 ± 6.1 months. All primary RLS patients and patients with gastrectomy responded to their third and fourth IV FCM treatments, and the responder rate for patients with anemia was 75.0% for the third FCM and 50.0% for the fourth FCM.

### 3.2. Comparison of Iron Characteristics in Patients with Primary RLS following Exposure to First and Second IV FCM

Table 3 summarizes changes in iron characteristics between the first and second IV FCM treatment in the primary RLS group (*n* = 26). There were no significant differences in pre-treatment hemoglobin level, or %TSAT between the first and second IV FCM treatments. However, serum ferritin before the second treatment was significantly higher than the baseline ferritin (before the first IV FCM) in both female and male subjects (Figure 2).

## 4. Discussion

In this study, we assessed the feasibility and efficacy of repeated treatment with IV FCM in patients with RLS who responded to an initial IV iron infusion. The response rate to repeated treatment in the primary RLS group was 80.8%, and the response rate to secondary RLS was 80.0% in patients with a history of gastrectomy and 81.8% in patients with anemia. Only two patients (3%) reported minor side effects (headache) after repeated infusion of FCM. A higher number of patients with primary RLS responded to repeated FCM than in previous randomized controlled studies investigating IV FCM treatment (45.8–59.4%) [4,13], and was even better than in the unblinded observational study of initial IV FCM infusion in our group (64.7%) [12]. To our knowledge, this study is the first to report the efficacy and safety of repeated dosing of IV FCM in RLS patients. A relatively large number of patients compared to a previous study of IV iron sucrose [15] and the use of IRLS, a validated assessment tool for RLS severity [20], reinforces the value of this study.

Data are limited to support the duration of IV iron treatment in RLS in the absence of long-term observational studies. The follow-up phase of two IV FCM randomized trials reported persistent treatment response in 60% of responders at 20 weeks, and 37.5% at 30 weeks, but these studies did not investigate the duration of treatment response by patients [4,13]. In a study that used supplemental IV iron sucrose mentioned above, the interval between initial treatment and the first supplemental dose ranged from 3 to 13 months, with a mean duration of 6.8 months (*n* = 5) [15]. In Earley’s study, the duration of IV iron dextran treatment effect in six responders ranged from 3 to 36 months, with a mean duration of 11.3 months [5]. In the present study, the time interval between initial IV FCM infusion and the first repeat ranged from 5 to 41 months, with a mean duration of 14.5 months in patients with primary RLS, which was longer than in previous reports. In terms of clinical practice, we could not provide recommendations on FCM administrations time intervals because durations of treatment effect were quite variable. Individualized therapeutic approaches would be needed, but 10–18 months could be used as a reference range, as about 70% of subjects had returning RLS symptoms within this period. The mean time intervals between treatments in patients with secondary RLS and history of gastrectomy and anemia were 10.6 and 11.6 months, respectively, which were not significantly different from those in the primary RLS group. It is tempting to speculate that the duration of IV iron treatment effect differs between patients with primary RLS and secondary RLS related to iron deficiency anemia or gastrectomy due to their different iron metabolism conditions. However, the sample size of patients with secondary RLS was too small to compare the duration of treatment effects with those of the primary RLS group.

In contrast to patients with secondary RLS with iron deficiency anemia or gastrectomy, who show favorable treatment responses to IV iron [12,21,22,23], the predictors of IV iron treatment response based on serum iron status or clinical factors is disputed in patients with primary RLS. Our previous retrospective open-label observational study demonstrated that low %TSAT was a predictor of good response to IV iron therapy [12] and a randomized controlled study showed a significant correlation between baseline %TSAT and improvement in IRLS [14]. However, most studies did not find a clear difference between responders and non-responders to IV iron in patients with RLS [4,13,19]. The present study suggests that a previous favorable response to IV iron treatment might be a strong predictor of patient response to repeated IV iron treatment in primary RLS, as well as secondary RLS. A similar favorable response to repeated FCM treatment between primary and secondary RLS also supported the recent concept of ‘RLS with comorbid conditions’, which consider secondary RLS may just result from a comorbid condition that exacerbates or triggers the physiologic mechanisms of RLS [24]. We found that serum ferritin was at a higher level before a second IV FCM treatment compared to baseline in the primary RLS group, in contrast to completely relapsed RLS symptoms (Figure 2). Moreover, these patients demonstrated favorable response to repeated iron supplement when their serum ferritin level was higher than the baseline, which suggests that serum ferritin does not reflect brain iron status after IV iron infusion, and, thus, it is not an appropriate biomarker of IV iron treatments efficacy in RLS. Serum ferritin is an important marker reflecting the iron status of the erythron [25]. Therefore, it is plausible that the equivalence between iron in the blood and body tissues, including the brain, after IV iron infusion might be altered. In contrast, % TSAT reflects the iron demand of the total body including brain, which was suggested as a probable predictor of treatment response in patients with primary RLS, as reported by our group previously [12]. In the present study, %TSAT before a second IV FCM was not higher than the baseline value, contrary to serum ferritin. However, it is difficult to suggest %TSAT as a biomarker to monitor the persistent effect of IV iron in RLS based on this study, in the absence of regular %TSAT monitoring during repeated IV FCM infusion.

Limitations of this study should be noted. First, it is a retrospective, open-label study. Second, there was lack of structured follow-up to assess the symptoms in all patients after FCM treatment, thus we do not know the exact time the symptoms re-occurred in the majority of patients. Future studies with long-term investigation of serial changes of iron status and RLS symptoms after IV iron treatment can facilitate the identification of an appropriate biomarker for monitoring persistent treatment effect of IV iron in RLS. Third, high efficacy in our study might be attributed to selection bias, as responders to initial IV FCM treatment were included in this study.

In summary, this study demonstrated the clinical efficacy and safety of repeated dosing with IV FCM in patients with primary and secondary RLS with compromised iron status. Among the initial responders whose symptoms relapsed after 5 to 41 months following initial treatment, more than 80% showed significant improvement in RLS after repeated 1000 mg IV FCM treatment. Despite the relapse of RLS symptoms, the patients’ serum ferritin levels remained at a higher level compared to baseline, suggesting inadequate measure of the iron demand of the brain. It is desirable to consider re-administration of IV FCM in patients with primary RLS, as well as secondary RLS patients showing compromised peripheral iron status, if they responded well to previous IV iron therapy and their symptoms recur. Further studies with a larger number of subjects and accurately designed randomized-controlled study are essential to identify RLS relapse biomarkers and to enable adequate monitoring after IV iron treatment in clinical practice.

## Figures and Tables

**Figure 1 jcm-11-01673-f001:**
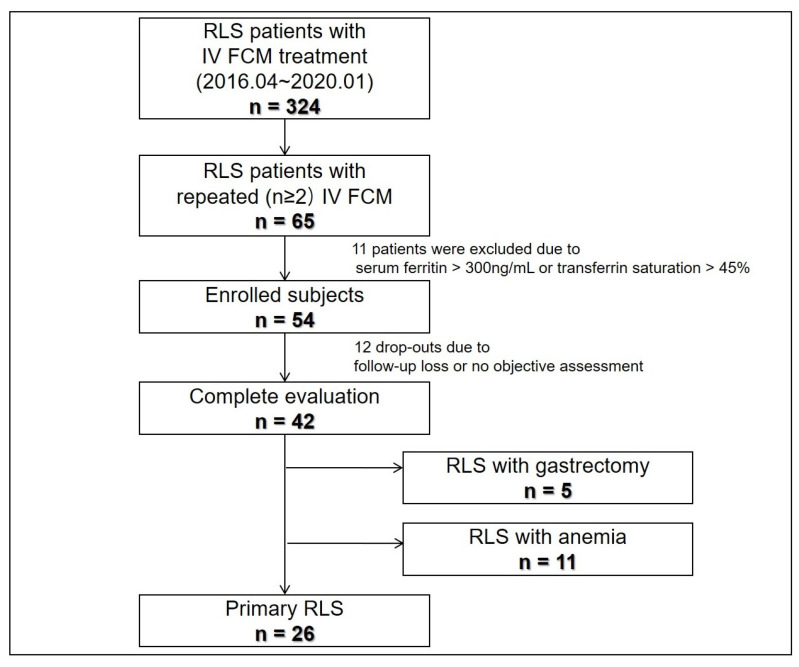
Study flow. FCM, ferric carboxymaltose; RLS, restless legs syndrome.

**Figure 2 jcm-11-01673-f002:**
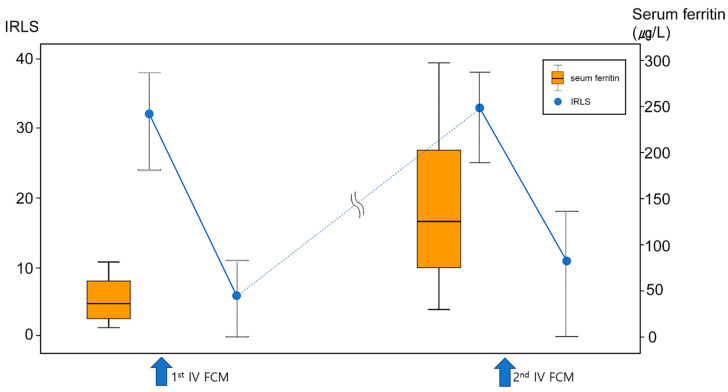
Changes in serum ferritin (orange box plot) and RLS symptoms (IRLS, blue dots) during repeated treatment of IV FCM in primary RLS group. Although IRLS completely returned to baseline (i.e., complete relapse) before second IV FCM, serum ferritin was at a higher level before a second treatment compared to baseline.

**Table 1 jcm-11-01673-t001:** Baseline characteristics of primary and secondary RLS patients.

	Primary (*n* = 26)	Gastrectomy (*n* = 5)	Anemia (*n* = 11)	*p*
Age, years	50.8 ± 18.9	68.6 ± 17.7	41.1 ± 16.2	0.028 *
Female, *n* (%)	22 (84.6)	3 (60.0)	11 (100.0)	0.067
Disease duration, years	11.2 ± 13.7	5.7 ± 8.1	6.9 ± 6.5	0.561
Pittsburgh sleep quality index	10.3 ± 4.7	10.5 ± 3.8	11.7 ± 4.2	0.777
Insomnia severity index	15.9 ± 7.0	19.5 ± 5.8	14.0 ± 7.7	0.502
HADS_Anxiety	8.0 ± 3.8	6.7 ± 5.5	11.6 ± 3.8	0.168
HADS_Depression	6.4 ± 4.7	9.7 ± 5.5	11.8 ± 2.0	0.061
Family history of RLS, *n* (%)	6 (23.1)	1 (20.0)	1 (9.1)	0.611
Previous RLS medication, *n* (%)	7 (26.9)	1 (20.0)	3 (27.3)	0.945
Baseline IRLS	30.5 ± 7.7	31.8 ± 4.7	27.8 ± 9.0	0.540
hemoglobin, g/dL	13.7 ± 1.0	11.0 ± 1.2	10.7 ± 1.5	<0.001 **
ferritin, ug/L	44.8 ± 28.7	12.9 ± 9.7	23.3 ± 28.3	0.020 ***
%TSAT	27.8 ± 10.2	6.2 ± 2.5	11.9 ± 7.7	<0.001 **

RLS, restless legs syndrome; HADS, hospital anxiety depression scale; IRLS, International restless legs scale score; %TSAT, percent transferrin saturation; Mean ± standard deviation. post hoc analysis; * gastrectomy > primary, anemia; ** primary > gastrectomy, anemia; *** no significant difference in post hoc analysis.

**Table 2 jcm-11-01673-t002:** The treatment response of repeated administration of FCM in primary and secondary RLS patients.

	Primary (*n* = 26)	Gastrectomy (*n* = 5)	Anemia (*n* = 11)	*p*
Number of FCM administration				0.131
twice, *n* (%)	24 (92.3)	4(80.0)	7 (63.6)	
Three times, *n* (%)	2 (7.7)	-	2 (18.2)	
Four times, *n* (%)	-	1 (20.0)	2 (18.2)	
Time interval between 1st–2nd FCM, months	14.5 ± 8.0	10.6 ± 3.2	11.6 ± 3.0	0.313
Responders, *n* (%)				N.A *
2nd treatment	22 (84.6)	4 (80.0)	9 (81.8)	
3rd treatment	2 (100)	1 (100)	3 (75)	
4th treatment	-	1 (100)	1 (50)	
IRLS change, % (after 2nd treatment)	62.2 (32.3)	73.9 (42.1)	64.9 (26.7)	0.760

FCM, ferric carboxymaltose; IRLS, International restless legs scale score * not available for chi-square tests due to limited number of subjects.

**Table 3 jcm-11-01673-t003:** Iron characteristics of repeated administration of IV FCM in primary RLS (*n* = 26; females = 22, males = 4).

	Before 1st Therapy	Before 2nd Therapy	*p* *
Female hemoglobin, g/dL	13.4 ± 0.8	13.2 ± 0.9	0.473
Male hemoglobin, g/dL	15.1 ± 0.6	15.1 ± 1.1	0.943
Female ferritin, μg/L	45.1 ± 30.9	147.5 ± 91.3	<0.001 ^†^
Male ferritin, μg/L	42.7 ± 13.0	174.3 ± 77.7	0.050 ^†^
Female %TSAT	26.7 ± 10.5	27.1 ± 7.2	0.942
Male %TSAT	35.1 ± 4.8	33.3 ± 5.4	0.195

FCM, ferric carboxymaltose; RLS, restless legs syndrome; %TSAT, percent transferrin saturation. * paired *t*-test ^†^
*p* ≤ 0.05.

## Data Availability

The datasets generated during and/or analyzed during the current study are available from the corresponding author on reasonable request.

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
