# Peer review of "Does Repeated Dosing of Intravenous Ferric Carboxymaltose Alleviate Symptoms of Restless Legs Syndrome?"

_jcm, 2022, doi:10.3390/jcm11061673_

Round 1
Reviewer 1 Report
This is an important investigation into the use of IV FCM for treating RLS patients. Many experts are now using IV iron therapy very often to treat RLS patients and further studies demonstrating the efficacy and information about duration of effect and re-administration are needed. This study provides valuable information on the use or IV FCM and repeated infusions.
A valuable and interesting observation from this study is noting that patients may have much higher ferritin levels after their first infusion (often way above the criteria for initial infusion) yet still benefit from a repeat infusion.
I have just a few comments. In the Introduction (line 39), experience amongst experts finds that the response to IV iron is often 4-8 weeks instead of 4-6 week.
The only other issue is the designation of primary versus secondary RLS. Many experts now consider that there is really no difference between primary and secondary RLS. Secondary RLS may just result from a comorbid condition (which is why we now refer to this as RLS with comorbid conditions) that exacerbates or triggers RLS. As such, both of those populations should respond quite similarly to IV iron (as was found in this study) with perhaps any difference attributed to initial lower ferritin/iron levels. A short discussion of these concepts may be helpful for readers of this study.
Reviewer 2 Report
The authors evaluated the effect of repeated administration of IV ferric carboxymaltose (FCM) in patients with restless legs syndrome (RLS) and found that approximately 80% of patients responded repeated FCM treatment (IRLS score <40%). Serum ferritin levels did not predict efficacy of treatment. Although this is a retrospective study, the study findings include important clinical implications. I have several comments on patient selection, RLS diagnosis and regular medical treatment between administration of FCM.
Introduction, line 57
To explore the benefit of repeated IV FCM treatment for patients with RLS diagnosed with relapsing RLS symptoms after a successful initial treatment with IV FCM,
-“patients with RLS diagnosed with relapsing RLS symptoms”is unclear and should be clarified.
Methods
The diagnostic method for RLS should be clearly stated, including diagnostic criteria.
Lines 93-94
In patients taking medications for RLS symptoms, IV FCM was administered without modification of 93 previous medication, and these patients were on a stable dosage of medications throughout the study duration
- I think it is difficult to keep the doses of other dopamine agonists and alpha2-delta ligand preparations consistent in a retrospective study; it should be specified whether the clinical application of FCM injections is limited to patients who are stable on other medications, or whether patients who did not have an increase or decrease in the dose of other oral medications were selected.
The interval between the first and second FCM administration was about one year. How were patients whose RLS symptoms worsened during this interval treated?
Are there any recommendations on FCM administration intervals?
Results, line 134
The authors described the time interval between the first and second FCM administration. Are the intervals between the second and third and third and fourth times similar?
In introduction or discussion section, it would be good to describe a little bit about the difference between FCM and oral iron, IV iron sucrose treatment.
Round 2
Reviewer 2 Report
The manuscript has been improved significantly and I have no further comments.